# Automated multiplex genome-scale engineering in yeast

Tong Si[1,2], Ran Chao[1,2], Yuhao Min[3], Yuying Wu[2], Wen Ren[2] & Huimin Zhao[1,2,3,4,5]

Genome-scale engineering is indispensable in understanding and engineering microorganisms, but the current tools are mainly limited to bacterial systems. Here we report an automated platform for multiplex genome-scale engineering in *Saccharomyces cerevisiae*, an important eukaryotic model and widely used microbial cell factory. Standardized genetic parts encoding overexpression and knockdown mutations of >90% yeast genes are created in a single step from a full-length cDNA library. With the aid of CRISPR-Cas, these genetic parts are iteratively integrated into the repetitive genomic sequences in a modular manner using robotic automation. This system allows functional mapping and multiplex optimization on a genome scale for diverse phenotypes including cellulase expression, isobutanol production, glycerol utilization and acetic acid tolerance, and may greatly accelerate future genome-scale engineering endeavours in yeast.

[1] Carl R. Woese Institute for Genomic Biology, University of Illinois at Urbana-Champaign, Urbana, Illinois 61801, USA. [2] Department of Chemical and Biomolecular Engineering, University of Illinois at Urbana-Champaign, Urbana, Illinois 61801, USA. [3] Department of Chemistry, University of Illinois at Urbana-Champaign, Urbana, Illinois 61801, USA. [4] Department of Biochemistry, University of Illinois at Urbana-Champaign, Urbana, Illinois 61801, USA. [5] Department of Bioengineering, University of Illinois at Urbana-Champaign, Urbana, Illinois 61801, USA. Correspondence and requests for materials should be addressed to H.Z. (email: zhao5@illinois.edu).

Microbial genome-scale engineering creates strain libraries for large-scale genotype–phenotype mapping[1], enabling important applications in fundamental biology[2], human diseases[3] and industrial biotechnology[4–6]. Given our limited understanding of complex cellular networks, it often requires both identification of genetic determinants and optimization of their expression in a concerted manner to improve target traits. However, most existing genome engineering methods perform these two tasks separately, by either modifying individual genes on a genome scale[2,4,5], or creating combinatorial diversity among pre-defined targets[6]. The main technical hurdle is the lack of a modular and expandable scheme to introduce genome-wide mutations in multiplex. Moreover, multiplex genome-scale modifications will result in enormous diversity, which requires robotic automation to facilitate creation and screening of genomic libraries, but very few examples have been reported to automate such efforts[6], and there is a general lack of standardized procedures of microbial genome-scale engineering.

To address these limitations, we sought to devise an automated system to integrate genome-wide screening and multiplex optimization. We first applied this system to *S. cerevisiae*, a well-established eukaryotic model with wide uses in fundamental research and industrial applications[7,8], as the current ability to reprogram the yeast genome is far behind bacterial hosts. In bacteria, efficient allelic replacement can be achieved using single-stranded DNAs with the help of bacteriophage proteins[9]. The so-called recombineering (recombination-based genetic engineering) technology enables genome-wide identification of genetic determinants using trackable multiplex recombineering (TRMR)[4] or combinatorial optimization using multiplex automated genome engineering (MAGE)[6] for a certain trait in *Escherichia coli*. However, recombineering only achieves an editing efficiency of 1% in *S. cerevisiae*[10] (compared with 30% in *E. coli*[6]), which is not sufficient to achieve genome-wide coverage or multiplex modifications when creating yeast libraries. Moreover, only short DNA oligonucleotides (90 nt) can be used in recombineering to introduce mutations efficiently without antibiotic selection[6]. When longer DNA cassettes were used, antibiotic selection became necessary[4]. This limitation on the editing scale is problematic in yeast in two ways: (1) genetic regulation is more complicated in eukaryotes, and small-nucleotide changes may be inefficient to modulate gene expression in yeast[11]; (2) for modifications on a larger scale, the use of antibiotic selection renders multiplex modifications difficult. Therefore, new methods other than recombineering must be developed for multiplex genome-scale engineering in yeast.

Our system integrated three major designs for automated yeast engineering: (1) To achieve genome-wide coverage, a normalized full-length-enriched complementary DNA (cDNA) library was used to construct genetic modulation parts encoding both overexpression and knockdown mutations. (2) Genetic modulation parts were constructed as donors for δ integration, an established method for multiplex integration into repetitive retrotransposon sequences in yeast[12–14]. (3) Integration efficiency was substantially improved via introduction of double-stranded breaks (DSBs) in the δ sequences, increasing the percentile of modified cells without the need for selection markers. On the basis of these designs, multiplex genome-wide mutations can be accumulated in a scalable manner using a standardized workflow, enabling automated genome-scale engineering in *S. cerevisiae*. Using this system, we not only performed genotype–phenotype mapping of both gain- and reduction-of-function mutations for >92% yeast genes but also successfully created multiplex diversity on a genome scale in a fully automated manner. A variety of industrially relevant traits were successfully improved in *S. cerevisiae*.

## Results

**Genome-wide overexpression and knockdown modulation parts.** We first sought to construct a collection of genetic modulation parts as a standardized method to create genome-scale mutations. Broadly, there are two main strategies to enable genome-wide perturbation, by directly editing the genome or introducing trans-acting regulatory elements[15]. Here we chose the latter for better scalability, as the same strategy (Fig. 1a) can be used to deliver a genome-scale library of regulatory elements to a cell population, and to deliver multiple regulatory elements targeting different genes in a single cell. Specifically, based on the SMART (Switching Mechanism At 5′-end of RNA Template) mechanism[16], we constructed a full-length-enriched cDNA library, which was further normalized via selective degradation of abundant cDNAs[17]. Two adaptor sequences were attached to the cDNA ends, facilitating directional insertion after a strong constitutive promoter $P_{TEF1}$ (Fig. 1b) on an episomal plasmid. Whereas genetic overexpression is achieved when full-length cDNA

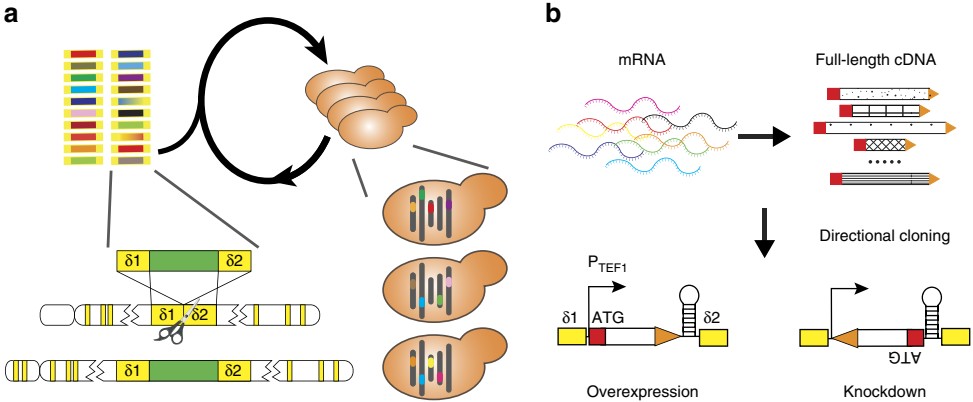

**Figure 1 | Scheme of automated genome-scale engineering in yeast.** (**a**) Design of a general method to create multiplex genomic mutations in yeast. Gene modulation parts encoding various genetic modifications were flanked by homologous δ sequences for iterative and multiplex integration into repetitive genomic sequences. To enable CRISPR-Cas for efficient and selection-free δ integration, a $P_{ADH2}$-Cas9 expression cassette was integrated into the CAD strain (a *S. cerevisiae* strain with a constitutive RNAi pathway[5]) to create the CAD-Cas9 strain. (**b**) Construction of a genome-wide modulation part library. Full-length-enriched cDNA library was directionally cloned after a constitutive promoter $P_{TEF1}$, and resulted in genetic overexpression or knockdown for sense and anti-sense configurations, respectively, when transformed into the CAD strain.

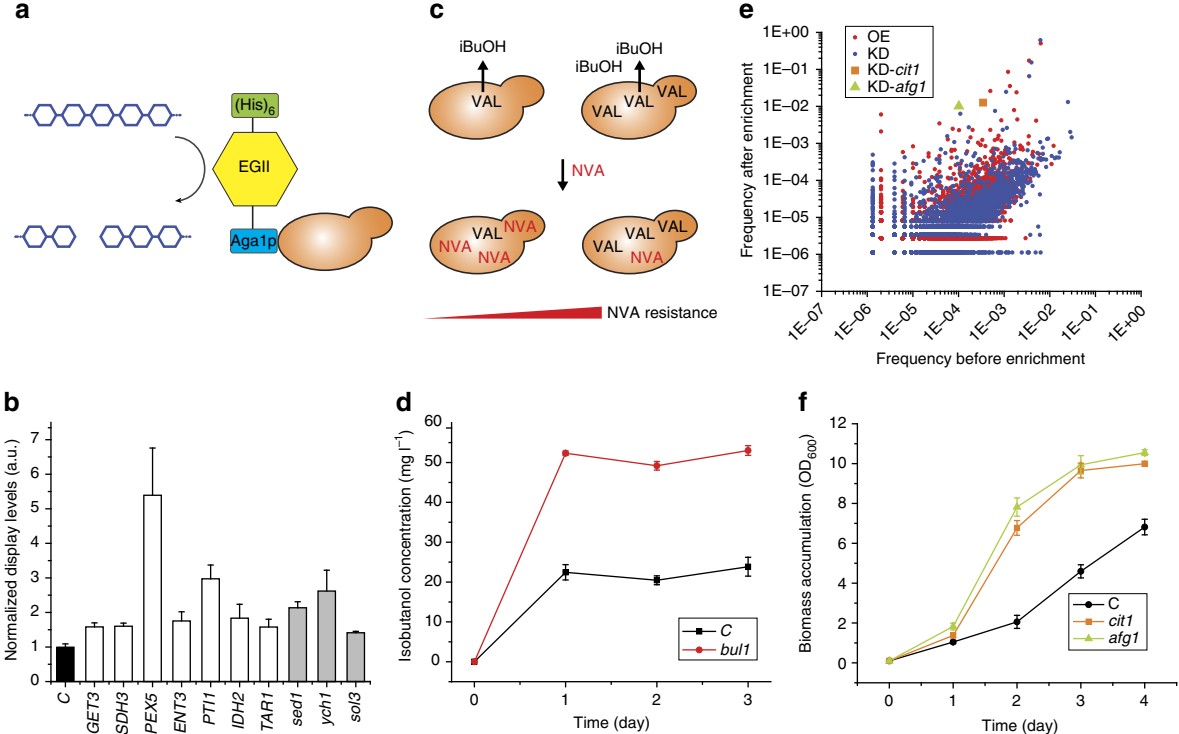

**Figure 2 | Genome-wide screening and gene–trait profiling using the gene modulation collection.** (**a**) Scheme of cell surface display of a cellulase (*Trichoderma reesei* endoglucanase II, EGII) in yeast. (**b**) Genetic mutations conferring improved EGII-display levels estimated using fluorescence immunostaining of a His-tag attached to EGII. Overexpression and knockdown targets are listed as white and grey bars, respectively. (**c**) Correlation of norvaline (NVA) resistance, valine (VAL) concentration and isobutanol (iBuOH) production. (**d**) A knockdown mutant with increased isobutanol production. (**e**) Frequencies of individual genetic part in strain libraries before and after serial transfer in synthetic media containing glycerol as the sole carbon source. Calculated from the NGS data, frequency = (read count of individual part + 0.5)/(total read count of all parts of the same modulation mode) (see Methods). Dots denote individual modulation part as red (overexpression part, OE), blue (knockdown part, KD), orange (KD for *cit1*) and blue (KD for *afg1*). (**f**) Genetic mutants conferring enhanced cell growth on glycerol. The same colour code is used as in **d**. *C* denotes the control strain (CAD-EGII in **b** and CAD in **d**,**f**) harbouring an empty pRS416 plasmid. Averages are defined as centre values, and error bars represent the mean ± s.d. from biological replicates ($n = 3$).

molecules are transcribed in the sense direction, genetic silencing is enabled via transcription of full-length anti-sense RNAs in a yeast strain integrated with a heterologous RNA interference (RNAi) pathway (the CAD strain)[5] (Fig. 1b). Next-generation sequencing (NGS) analysis indicated that ~92.3% of all endogenous yeast genes were included in the library (Supplementary Data 1), and 19 out of 20 randomly picked plasmids contained full-length open reading frames for either overexpression or knockdown configurations (Supplementary Table 1).

**Genome-wide screening and functional profiling.** We applied this collection for genome-wide screening to engineer a range of industrially relevant phenotypes, including protein secretion, chemical tolerance/production and substrate utilization. To ensure adequate coverage, $>10^6$ independent clones were obtained for each strain library. Retransformation into a fresh strain background was performed for isolated genetic parts to confirm their impact on a target trait. We first tried to improve cell surface display of a cellulase (Fig. 2a), a highly desirable trait for lignocellulosic biofuel production[18,19]. A His-tag was attached to the N-terminus of the cellulase to allow display-level quantification using immunostaining[20] (Fig. 2a), and both up- and downregulation mutations were identified to enhance cellulase-displaying levels (Fig. 2b) and hydrolysis rates of a cellulose substrate (Supplementary Fig. 1). Moreover, we screened

for improved production of isobutanol as a second-generation biofuel. Enhanced resistance to norvaline, a toxic analogue of valine as a precursor for isobutanol, has been linked to isobutanol overproduction[21] (Fig. 2c). Following serial transfer in norvaline-containing media (Supplementary Table 2), a knockdown mutation conferring improved isobutanol titres was identified (Fig. 2d).

We further conducted genome-wide mapping of gene–trait relations. We targeted substrate utilization of glycerol, a promising renewable feedstock for microbial fermentation[22]. Plasmid DNA was isolated from yeast libraries before and after serial transfer in glycerol media (Supplementary Table 2), and NGS analysis was used to track frequency changes of individual genetic part (Fig. 2e; Supplementary Data 1 and 2). Consistent with previous findings[23], functional clustering of genes targeted by enriched parts indicated the important roles of the tricarboxylic acid (TCA) pathway and electron transport in aerobic respiration to glycerol utilization (Supplementary Data 2). Two knockdown parts with substantially elevated frequencies (fitness) were also isolated by streaking individual clones from the enriched library (Fig. 2e), and improved cell growth on glycerol was confirmed after retransformation (Fig. 2f).

For all three phenotypes, gene silencing of the isolated knockdown mutations was confirmed using real-time quantitative PCR (qPCR; Supplementary Fig. 2). For identified genetic mutations, it is possible to speculate the mechanisms underlying the improved phenotypes from known gene functions

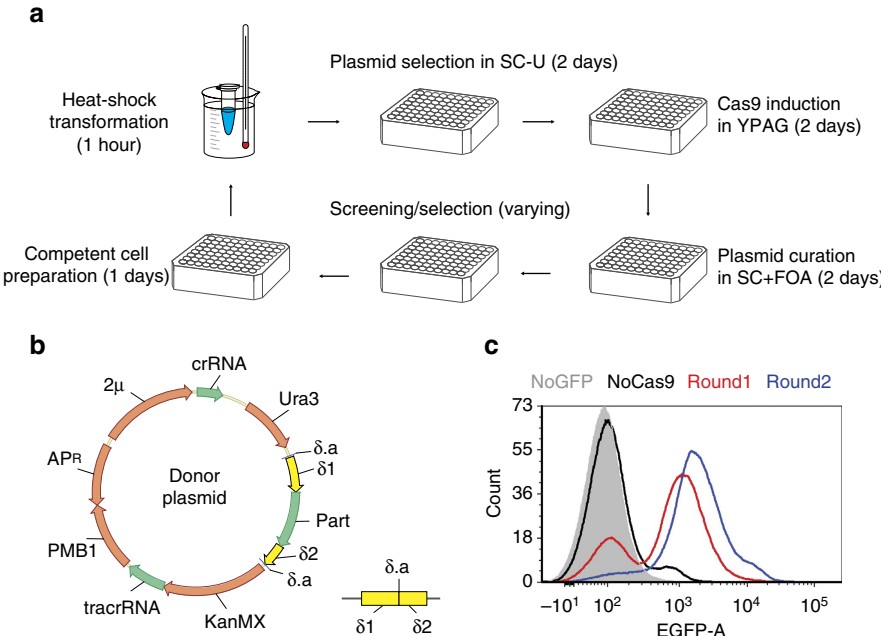

**Figure 3 | CRISPR-assisted δ integration of a GFP reporter.** (**a**) Workflow for iterative and multiplex integration. A screening/selection step can be included after plasmid curation to enrich mutants with desirable traits. (**b**) Donor plasmid of CRISPR-assisted δ integration, consisting of a 2 μ high-copy replication origin, CRISPR RNAs to direct Cas9 to the δ.a site in both the donor and genome, the selection markers (*KanMX* and *URA3*) for plasmid maintenance and curation, and an integration donor cassette flanked by homologous δ sequences (δ1 and δ2). (**c**) Genomic accumulation of GFP donors. Flow cytometry histogram overlays depict GFP fluorescence of cell populations after one (red) or two (blue) rounds of integration with the GFP donor plasmid in CAD-Cas9. Negative control (grey shade, NoGFP) was treated with an empty pRS426 plasmid in CAD-Cas9. CAD treated with the GFP donor in two rounds of integration was denoted as NoCas9 (black). In **c**, representative histograms are presented from biological triplicates.

(Supplementary Table 3). For instance, the *SED1* gene (Fig. 2b) encodes a stress-induced structural cell wall protein Sed1p with high display levels[24], and has been isolated previously from genome-wide screening to increase cell surface display of heterologous proteins in *S. cerevisiae*[25]. It is possible that suppressed expression of Sed1p can help to accommodate heterologous proteins that compete for display capacity of the cell surface. Moreover, increased isobutanol titres have been observed with reduced expression of the general amino-acid permease Gap1p (ref. 26), whose intracellular trafficking is regulated by Bul1p (ref. 27). Also, the knockdown cassette of *CIT1* (Fig. 2f) targets citrate synthase as a rate-limiting enzyme of the TCA cycle, and reduced TCA activity has been observed in fast glycerol-metabolizing *E. coli*[28]. While identification of known genetic targets for relevant phenotypes suggests that the part collection is effective for genome-wide profiling, further research is needed for functional elucidation.

**Highly efficient multiplex integration enabled by CRISPR.** Next, we tried to establish a standard workflow to incorporate genetic modulation parts in the yeast genome. We sought to improve δ integration efficiency by creating DSBs in the genomic δ sequences using the CRISPR-Cas system[29] (Figs 1a and 3a,b). Modified from our previous design[30,31], CRISPR-component configuration (Fig. 3b) and process parameters (Fig. 3a) were optimized to increase the percentile of integrated cells in a single round of transformation (Supplementary Fig. 3). In particular, the type II *Streptococcus pyogenes* Cas9 gene was integrated into the yeast genome (the CAD-Cas9 strain), and a glucose-repressed promoter $P_{ADH2}$ was used to drive inducible expression of Cas9. To improve recombination rates, the

integration cassette flanked by homologous δ sequences was cloned in a high-copy donor plasmid, which also contained the targeting CRISPR RNA and trans-activating CRISPR RNA cassettes to direct Cas9 to the δ sites (Fig. 3b). The integration workflow included three steps: transformation of the yeast cells with the donor library, induction of Cas9 expression to stimulate integration and plasmid curation by counter-selection (Fig. 3a). After plasmid curation, cells were ready for screening or the next round of integration. Following the optimized workflow (Fig. 3a), when a green fluorescent protein (GFP) expression cassette was used as the integration donor, ∼70 and 90% of the cell population exhibited GFP fluorescence after the first and second rounds of integration (Fig. 3c; Supplementary Fig. 3h,i). Only ∼0.1% GFP-positive cells observed following the same workflow without Cas9 (Fig. 3c; Supplementary Fig. 3j), indicating the necessity for DSB generation for stimulating integration and effective curation of donor plasmids. After the first round of integration, 13 out of 20 (65%) randomly isolated individual clones showed GFP fluorescence (Supplementary Fig. 4a), consistent with the percentile (70%) of GFP-positive cells observed on the population level (Supplementary Fig. 3h). Together with the top three brightest clones isolated using fluorescence-activated cell sorting (FACS), the 23 clones exhibited a 20-fold variation of mean GFP fluorescence intensities (Supplementary Fig. 4a–c), and the phenotypic diversity may be resulted from differences in copy numbers (Supplementary Fig. 4a) and genomic loci of integration among engineered strains. Also, as 30% of the cells remained unmodified after one round of integration, these cells can serve as built-in controls for the wild-type parent and spontaneous adaptive mutants during screening.

Next, we manually tested the feasibility of multiplex genome-scale engineering scheme (Fig. 1). Using genome-wide

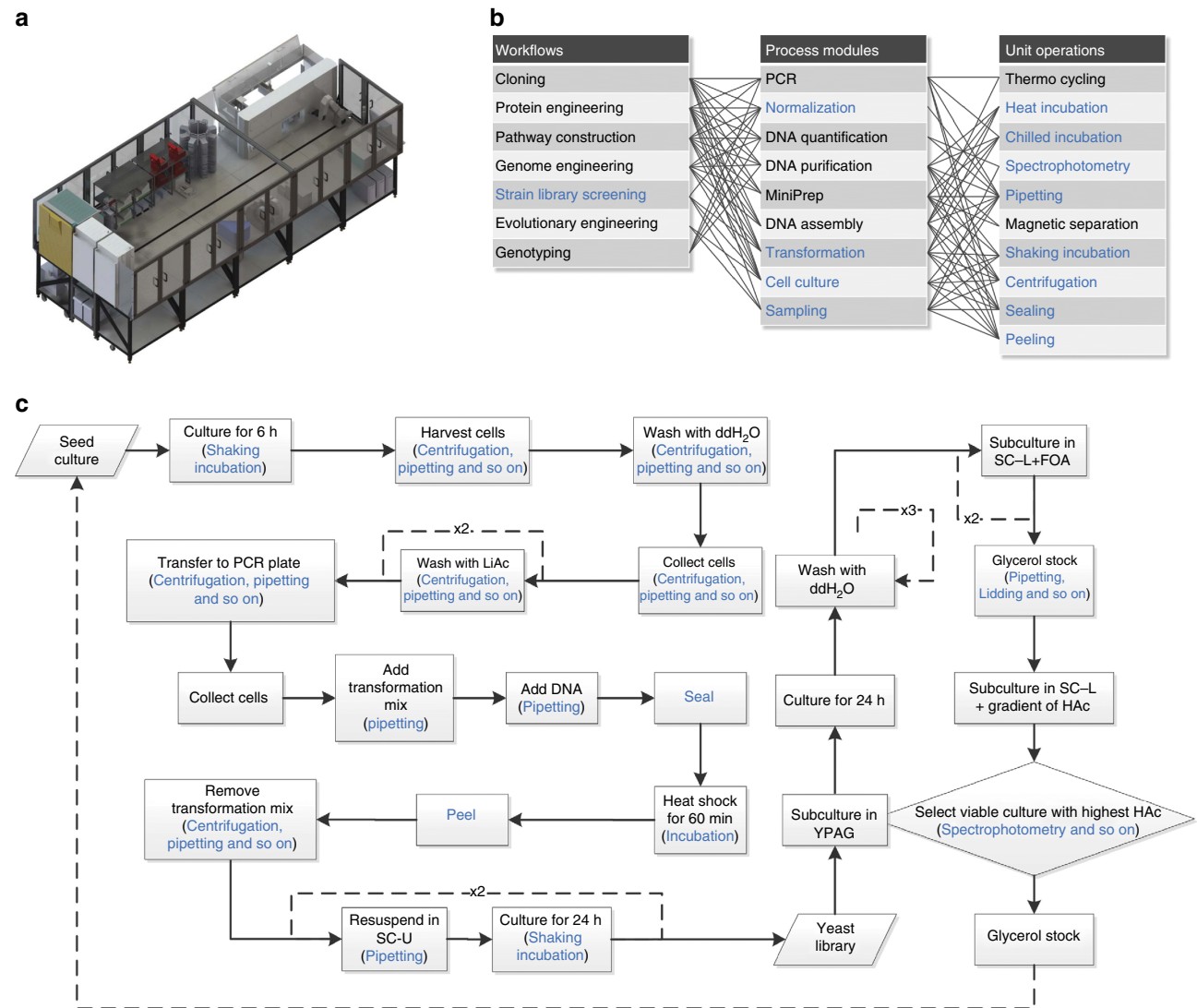

**Figure 4 | Automated yeast strain engineering using iBioFAB.** (**a**) Hardware layout of iBioFAB (adapted with permission from ref. 32. Copyright (2017) American Chemical Society). (**b**) Process modules and unit operations in the workflow of yeast engineering. Any subset of all the unit operations enabled by iBioFAB can be programmed in custom-designed sequences to perform necessary process modules for creating and screening strain libraries in an iterative and automated manner. (**c**) Process flow diagram. Unit operations used in the yeast engineering workflow are marked in blue in **b,c**.

modulation collection as integration donors, genetic parts targeting 5,051 yeast genes were incorporated in the genomes of a strain library after one round of integration (80.6% genomic coverage, Supplementary Data 1c). We randomly isolated five yeast colonies streaked from the integration library, and observed that 3–10 different integrated parts were recovered from each genome (Supplementary Data 1d). These observations suggest that our method is effective in both creating genome-wide mutations in a cell population and introducing multiplex modifications in a single cell. We then performed three iterative rounds of integration (Fig. 3a), and spread the resultant cell culture on agar plates containing glycerol as the sole carbon source. Two mutants (G1* and G6) were isolated with improved cell growth on glycerol (Supplementary Fig. 5), and genomic sequencing results indicated there were 44 and 19 different modulation parts integrated in the G1* and G6 mutants, respectively (Supplementary Table 4). This result indicates our genome-scale engineering workflow is able to create phenotypic diversities via multiplex integration of genetic modulation parts.

**Automated multiplex genome-scale engineering in yeast**. We next sought to automate the genome-scale engineering workflow using a biological foundry—Illinois Biological Foundry for Advanced Bioengineering (iBioFAB, Fig. 4a), which is an integrated platform for automated biomanufacturing consisting of component instruments, a central robotic platform and a modular computational framework[32]. We previously employed iBioFAB for automated synthesis of transcription activator-like effector nucleases, and the modular design of iBioFAB allows facile reconfiguration for other workflows on the same platform. Briefly, a new workflow can be translated into an executable sequence of unit operations enabled by the component instruments, such as liquid handling, centrifugation, incubation and $OD_{600}$ measurement (Fig. 4b). The reprogrammed operation sequence is then orchestrated using the central computational framework and the integrated robotic system[32]. Compared with the transcription activator-like effector nuclease synthesis workflow, a new process module for yeast cell transformation was developed based on heat-shock protocol[33] (Fig. 4c). In each batch, this process module can perform 192 parallel

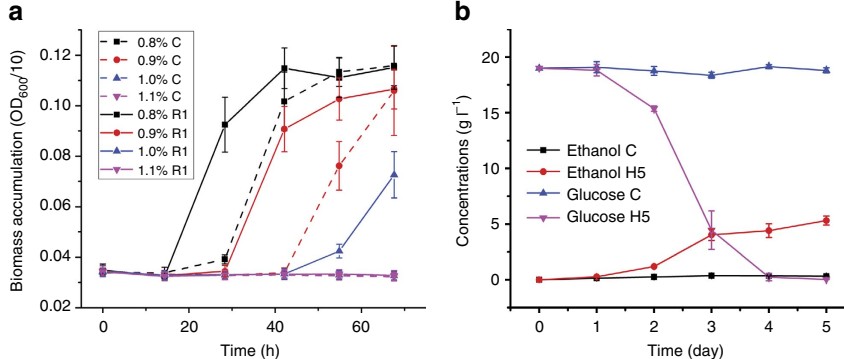

**Figure 5 | Screening of HAc-resistant yeast strains.** (**a**) Improved HAc tolerance in a strain library after the first round of integration (R1, solid line) compared with the wild-type control strain (dashed line) under various levels of HAc stress. (**b**) Ethanol fermentation with 1.1% (v/v) HAc by the parent and a mutant yeast strain isolated using iBioFAB. In **a**, error bars represent the mean ± s.d. from technical replicates ($n = 16$), where the cell libraries after plasmid curation were combined before being inoculated into 16 microtitre wells of selection media for each HAc concentration. In **b**, error bars represent the mean ± s.d. from biological rplicates ($n = 3$).

transformation reactions at an efficiency of $10^4 \mu g^{-1}$ of plasmid DNA to generate $> 10^6$ independent transformants. An online spectrophotometric analysis module was also developed for monitoring cell growth (Fig. 4c), and $10^9$ and $10^8$ cells can be transferred between plates at inoculum ratios of 10 and 1% for subculturing and screening, respectively.

Here we chose acetic acid (HAc) tolerance, a highly desirable trait in lignocellulosic ethanol fermentation[34], as the target phenotype. A screening step was included in the integration workflow (Fig. 4c), where the strain library after each round of integration was selected in synthetic media with a gradient of HAc stresses ranging from 0.8 to 1.1% (v/v; Supplementary Fig. 6). In the first round (R1), we observed substantial improvement in HAc tolerance of the strain library relative to a wild-type control strain harbouring an empty donor plasmid lacking CRISPR RNAs (Fig. 5a). In the first, second and third rounds (R1, R2 and R3), cells were selected from 0.9, 1.0 and 1.1% HAc media, respectively (Supplementary Fig. 6a) for further rounds of engineering, and a general trend of R3 > R2 ∼ R1 > C was observed for biomass accumulation under different HAc concentrations (Supplementary Fig. 6b–d). For the parent strain (CAD-Cas9), on the contrary, we observed no substantial biomass accumulation in 1.0 and 1.1% HAc media during three rounds of serial transfer under the same selective pressures, although cell growth was improved in the presence of 0.8 and 0.9% HAc (Supplementary Fig. 7a). These observations suggest that the occurrence of HAc resistance is greatly accelerated using our workflow relative to traditional evolutionary engineering. NGS analysis of integrated parts in the genomes of selected populations indicated dynamic inclusion and elimination of the modulation parts (Supplementary Fig. 6e; Supplementary Data 3a), which may be resulted from genomic introduction of part donors in the integration step and counter-selection of less resistant cells with increasing HAc stresses, respectively. For the integrated parts, targeted genes were enriched in functional classes that are known to be related to HAc resistance (Supplementary Fig. 8). In particular, 86 targeted genes were previously identified as genetic determinants of HAc tolerance (Supplementary Data 3a,b).

After the third round, we randomly isolated five individual clones (H1–H5) from the selected population after streaking on agar plates, and compared their HAc resistance with the parent strain (CAD-Cas) before and after traditional adaptive evolution. In terms of biomass accumulation, HAc resistance levels were in the order of engineered mutants > adaptive

population > parent in 0.9% HAc media (Supplementary Fig. 7b). In the presence of 1.1% HAc, a condition that completely inhibited cell growth for the parent strain and adaptive population, four engineered mutants acquired the capability to grow (Supplementary Fig. 7c). We then examined the stability of the engineered mutants. After 100 generations of cell division in non-selective media, four out of five mutant strains showed no substantial differences in biomass accumulation in 0.9 and 1.1% HAc media (Supplementary Fig. 7d). However, H3 lost the capability to grow in 1.1% HAc media after long-term cultivation (Supplementary Fig. 7d). These results are consistent with previous reports on the general stability and possibility of genetic instability of δ-integration strains[13,35], and the instability may be resulted from homologous recombination among the same promoter and terminator sequences in multiple integrated cassettes.

Finally, we performed flask fermentation using the H5 mutant, which showed the highest HAc resistance among isolated strains. While H5 converted $20 g l^{-1}$ glucose into $5.9 g l^{-1}$ ethanol in 4 days in the presence of 1.1% HAc, the parent strain exhibited no glucose consumption or ethanol production (Fig. 5b). To our knowledge, such HAc resistance is the highest level reported in the literature (Supplementary Data 3c), which was achieved within ∼1 month (Fig. 3a) instead of much longer time periods for adaptive evolution or rational engineering. This achievement indicates our method greatly accelerates the emergence of desirable phenotypes, via more effective creation and screening of the genomic diversity. Genome sequencing revealed that at least 26 different modulation parts were integrated in the H5 genome (Supplementary Table 4), suggesting that this strategy is effective in creating multiplex diversity. However, further research is needed to understand the underlying mechanisms of the resistance phenotype.

## Discussion

In this study, we developed a framework for modular introduction of multiplex genome-wide modifications in yeast, which presents three major advances. First, simply cloning from cDNA libraries, both overexpression and knockdown mutations can be created on a genome scale, which is proved effective in studying various traits including protein and chemical production, substrate utilization and inhibitor tolerance. Compared with existing yeast genome-scale tools that only permit a single modulation mode (upregulation[36], downregulation[5,37] or

knockout[2]), the current method creates a more comprehensive diversity space by allowing genetic overexpression and silencing in a single library. Also, as gene deletion[30] and large pathway integration[31] can also be generated using episomal vectors harbouring CRISPR components and homologous recombination donors, it is potentially possible to include these mutation types in the current system as well by modifying the donor plasmid configuration. In *E. coli*, recombineering enables insertion of synthetic DNA cassettes upstream of >95% genes to achieve both up- and downregulation (the TRMR method[4]) with antibiotic selection. This strategy[4], however, cannot be readily used in *S. cerevisiae*, mainly due to the low efficiency of recombineering (1% recombination efficiency in yeast using YOGE[10], compared with 30% in *E. coli* using MAGE[6], and 70% using our method). Also, genetic regulation in eukaryotes is more complicated than that in bacteria. For example, to create a variant library with 8–120% transcription activities compared to the wild-type yeast $P_{TEF1}$ promoter, introduction of 4–71 mutations randomly distributed in the 401 bp of promoter sequence was needed[11]. Hence, it is difficult to generate well-defined overexpression or knockdown mutations by inserting short DNA sequences in the yeast promoter regions as achieved in bacteria using TRMR[4]. Moreover, TRMR requires a sophisticated protocol to generate the barcoded pools of synthetic cassettes from microarray oligonucleotides[4], while our method uses routine DNA cloning techniques to prepare the library, and gene targets can be readily identified from part sequences using NGS analysis without barcodes (Supplementary Fig. 9). Nonetheless, although our library covers >92% yeast genes, cDNA-derived libraries may suffer from poor representativeness, due to no or low expression of certain genes under a given condition, as well as incomplete cDNA sequences (critical for genetic overexpression). Therefore, it is essential to obtain high-quality cDNA library (full-length enriched, normalized and combined from different culture conditions and so on).

Second, gene expression modulation is achieved without modifying target genomic sequences, so that diverse mutations can be introduced in a modular and scalable manner as standardized integration donors. In *E. coli*, recombineering enables combinatorial optimization of 24 pre-defined gene targets using the MAGE method[6]. When creating a genome-scale library using recombineering, however, it still requires antibiotic selection to enrich modified cells (TRMR[4]), which renders creation of multiplex mutations difficult. To achieve multiplex genome-scale engineering in *E. coli*, a two-step strategy has been developed, whereby a pool of genetic determinants of a target trait was first identified from genome-wide screening using TRMR, and then combinatorial optimization within this gene pool is performed using MAGE[38]. However, individual beneficial mutations are not necessarily addictive for a given trait[5,38], and genome-wide search may be essential to identify genetic interactions for synergistic beneficial mutations[5,39]. Therefore, we argue that it is necessary to integrate the genome-wide screening and multiplex optimization in a single method. Notably, TRMR[4] and MAGE[6] apply different strategies to modulate gene expression—TRMR inserts synthetic modulation cassettes in the promoter regions, while MAGE introduces nucleotide-level mutations in the RBSs. As a result, these two methods cannot be seamlessly integrated[38]. On the contrary, using trans-acting modulation cassettes and modular δ integration, our system allows continuous introduction of diverse genetic mutations in multiplex following the same procedure, which is critical to achieve automated yeast engineering. Broadly, genome editing[2,4,6,10,30,40] and trans-acting regulatory elements[5,41] are two very important strategies for genome-scale engineering and combinatorial optimization in microorganisms[15]. Given that few genome-editing tools

have demonstrated the capacity for combinatorial genome-scale engineering in yeast, here we chose to use trans-acting genetic modulation parts, as it is highly efficient to deliver a large number of parts for genomic coverage, and to introduce many parts in a single cell for multiplex mutations. In the future, CRISPR may have the potential considering the high editing efficiency, the use for multiplex editing[30] and diverse modulation modes[42] in yeast, and the application in genome-wide screening in human cells[43,44], but further development is still needed.

Third, a defined process was designed for iterative execution by automated robotic systems. Automation is essential as combinatorial genome-scale engineering often generates huge diversities. Theoretically, multiplex ($N$) integration of genome-wide modulation parts ($\sim 10^4$ modifications covering overexpression and knockdown of $\sim 6,000$ yeast genes) can potentially create a gigantic number of mutants ($10^{4N}$). Although this diversity far exceeds the actual sizes of cell populations at any given time and therefore cannot be comprehensively screened, it can be collectively created through successive mutagenesis cycles, as demonstrated in this study and elsewhere previously[6]. Still, even such enormous diversity represents only a subset of all possible genomic variants. Therefore, automation is critical to accelerate and scale up creation and screening of strain libraries[6]. This work also demonstrates the importance of synthetic biology principles in achieving automation, including standardization and modularization.

On the other hand, further improvement is possible to address certain limitations. It is inherently challenging to speculate the mechanisms conferring improved phenotypes with multiple mutations in a single cell. While many approaches are developed to study mutants with many modifications (for example, mutants isolated from adaptive evolution experiments[45]), here we propose three solutions based on our design: (1) integrated genetic parts can be recovered using PCR and sub-cloned into a plasmid, and the impact of individual mutations on a target trait can be examined; (2) the recovered parts from an isolated mutant can be used as integration donors to a fresh cell population, so that combinations of the recovered parts can be generated to differentiate beneficial mutations from neutral or deleterious hitchhikers, and to identify synergistic mutations; (3) here we purposely maximized integration efficiency to explore the potential of creating combinatorial diversity, but it is possible to reduce integration copies by targeting less abundant repetitive genomic elements, or fine-tuning expression of CRISPR components (Supplementary Fig. 3). The third practice may also help to address some potential complications, including chromosomal rearrangement due to massive cleavage at the repetitive genomic sequences (Supplementary Fig. 10), as well as saturation of the RNAi machinery in the presence of many knockdown parts. Moreover, although mutant strains with improved traits can be quickly isolated, they are not considered ideal for industrial applications due to deleterious hitchhiker mutations, a limited modulation range on gene expression (with no gene deletion) and possible genetic instability as observed for H3 (Supplementary Fig. 7d). Therefore, we consider our method primarily as a genome-scale discovery tool to search and understand genetic basis, especially multiplex mutations, for obtaining desirable phenotypes.

In conclusion, we developed a platform for automated multiplex genome-scale engineering in *S. cerevisiae*. The individual components of this platform—trackable genome-wide modulation parts, CRISPR-assisted δ integration and automated workflows using iBioFAB—can be performed in isolation or combination as needed, and we envision this flexible system can advance a wide range of research in *S. cerevisiae*. In addition, this multiplex genome-scale engineering design

combining standardized modulation parts and combinatorial integration may potentially be extended to other microbial hosts.

## Methods

**Strains and cultivation conditions.** *S. cerevisiae* strain CEN.PK2-1c (*MATa ura3-52 trp1-289 leu2-3,112 his3Δ1 MAL2-8C SUC2*) was purchased from EUROSCARF (Frankfurt, Germany). The CAD strain was constructed previously[5] via integration of an RNAi pathway into the CEN.PK2-1c genome. Zymo 5α Z-competent *E. coli* (Zymo Research, Irvine, CA) and NEB 10β Electrocompetent *E. coli* (New England Biolabs, Ipswich, MA) were used for plasmid amplification and library construction, respectively. *S. cerevisiae* strains were cultivated in either synthetic complete (SC) dropout medium (0.17% Difco yeast nitrogen base without amino acids and ammonium sulfate, 0.5% ammonium sulfate and 0.083% amino-acid dropout mix, 0.01% adenine hemisulfate and 2% glucose) or YPAD medium (1% yeast extract, 2% peptone, 0.01% adenine hemisulfate and 2% glucose). For induction of Cas9 expression, 2% ethanol or 2% galactose was used instead of 2% glucose in the YPA medium to create YPAE or YPAG media, respectively. The induction media were supplemented with $1 \, g \, l^{-1}$ G418 to maintain the donor plasmid. For plasmid curation, $1 \, g \, l^{-1}$ 5-fluoroorotic acid (5-FOA) was added into the SC medium (pH = 4). *S. cerevisiae* strains were cultured at 30 °C and with 250 r.p.m. agitation in baffled shake-flasks for aerobic growth, and at 30 °C and 100 r.p.m. in un-baffled shake-flasks for oxygen-limited fermentation. *E. coli* strains were cultured at 37 °C and 250 r.p.m. in the Luria broth medium (Fisher Scientific, Pittsburgh, PA) with the supplement of $100 \, \mu g \, ml^{-1}$ ampicillin. All chemicals were purchased through Sigma-Aldrich or Fisher Scientific.

**DNA manipulation.** Plasmid cloning was performed using the Gibson Assembly Cloning kit from New England Biolabs following the manufacturer's instructions or using the DNA assembler method[46]. The list of primers used can be found in Supplementary Table 5. For DNA manipulations, yeast plasmids were isolated using a Zymoprep II yeast plasmid isolation kit (Zymo Research) and transferred into *E. coli* for amplification. QIAprep Spin Plasmid Mini-prep kits (Qiagen, Valencia, CA) were employed to prepare plasmid DNA from *E. coli*. All enzymes used for recombinant DNA cloning were from New England Biolabs unless otherwise noted. PCR, digestion and ligation products were purified by QIAquick PCR Purification and Gel Extraction kits (Qiagen). Genomic DNA isolation was performed using the YeaStar Genomic DNA kit (Zymo Research).

**Construction of a genome-wide modulation library from cDNA.** As both genetic overexpression and knockdown screens are important for genotype–phenotype mapping, it is desirable to incorporate both types of genetic modifications simultaneously to accelerate the screen process. Inspired by our previous observation that full-length anti-sense RNAs can efficiently knockdown a target gene through RNAi in yeast[5], we devised a method for one-step construction of a comprehensive genome-wide library through directional cloning of full-length cDNAs. When a full-length cDNA molecule is inserted in the sense direction under the control of a constitutive promoter, the resultant construct will lead to gene overexpression of the contained open reading frame (Fig. 1b). When inserted in the reversed direction, on the other hand, the resultant construct will transcribe an anti-sense RNA of the target gene, and elicit gene silencing in the presence of RNAi machinery[5] (Fig. 1b). Directional cloning can be achieved by adding two different adaptor sequences to the ends of a cDNA molecule, and thus insertion directions can be controlled by arranging the homologous adaptor sequences in a desired order in an expression cassette (Fig. 1b).

We first synthesized a normalized full-length cDNA library, whereby two 15 bp adaptor sequences, 5′-AAGCAGTGGTATCAA-3′ and 5′-CGGGGTACGAT GAGA-3′, were incorporated at the 5′ and 3′ ends of the cDNA molecules, respectively. Briefly, total RNAs from the overnight culture of the CEN.PK2-1c strain in the YPAD medium were isolated using the RNeasy mini kit (Qiagen). A cDNA library was synthesized using the In-Fusion SMARTer Directional cDNA Library Construction kit (Clontech Laboratories, Mountain View, CA) with some modifications. In particular, the double-stranded cDNA library was treated using the Trimmer-2 cDNA normalization kit (Evrogen, Moscow, Russia) according to the manufacturer's instructions, whereby selective degradation of abundant cDNAs can be achieved using a duplex-specific nuclease from the Kamchatka crab[17]. Smaller cDNA fragments were then removed using the size-fractionation step included in the SMARTer kit. It has been reported that normalization and size-fractionation steps remarkably improve quality and representativeness of the full-length cDNA libraries[47].

For directional cloning, we developed a constitutive expression cassette consisting of a $P_{TEF1}$ promoter and a $T_{PGK1}$ terminator (Fig. 1b). The treated cDNA library was cloned into the linearized backbone vectors using the Gibson Assembly Cloning kit (New England Biolabs). The assembled product was purified using *n*-butanol precipitation and resuspended in 10 μl of double-distilled water (ddH$_2$O). The purified product was transformed via electroporation into the NEB 10β electrocompetent *E. coli* strain (*ccdB* sensitive). Following recovery in 2 ml of the SOC medium for 1 h, the cells were spread on 20 Luria broth + ampicillin plates. Cultivation on solid media can minimize uneven

amplification of clones containing varying sizes of plasmids. More than $10^6$ independent *E. coli* colonies were obtained for both overexpression and knockdown libraries, representing > 100-fold redundancy of ∼6,000 yeast genes.

Ten colonies were randomly picked from each *E. coli* library (overexpression and knockdown), and the isolated plasmids were analysed via DNA sequencing. The results indicated that all the cDNAs were inserted in the expected directions, and most insets (19/20) were full-length cDNAs (Supplementary Table 1). The only partial cDNA had the characteristic poly-A tail in an mRNA molecule, possibly synthesized from a premature transcriptional intermediate. To estimate genomic coverage, 50 μg of plasmid DNA from both overexpression and knockdown libraries (100 μg in total) was transformed into the CAD strain using heat-shock transformation. After amplified on SC-U plates for 4 days, transformants were collected into 15 ml of ddH$_2$O and 1 ml of cell suspension was used to isolate yeast plasmid DNA. Modulation parts were PCR-amplified using PRO index primer F (806rcbc930) and PRO universal primer R, as well as TER index primer F (806rcbc930) and TER universal primer R. PCR products were subjected to NGS analysis as detailed below.

**Screening of EGII-displaying mutants.** The expression cassette of *Trichoderma reesei* endoglucanase II (EGII, EC 3.2.1.4) was derived from a previous work in our lab with some modifications[20]. In particular, the prepro secretion signal peptide, the His epitope tag and the EGII open reading frame were kept intact and amplified from the original cassette $P_{GAL10}$-(prepro signal peptide)-His-EGII-docS-$T_{ADH1}$ using primers PR1001/PR1002. The dockerin module docS was replaced with the a-agglutinin mating adhesion receptor (amplified from yeast genomic DNA using primers PR1003/PR1004) to allow direct anchoring of EGII on the cell surface. The promoter and terminator were changed to $P_{TEF1}$ and $T_{PGK1}$, respectively, to permit constitutive expression. The cellulase-displaying strain, CAD-EGII, was constructed by integrating the EGII-displaying cassette into the *leu2* site of the CEN.PK2-1c genome using pRS405. Plasmid DNA of the modulation part library constructed on pRS416 was used to transform the CAD-EGII strain by the standard LiAc/ssDNA/PEG protocol[33] with an optimized condition, where 100 μg plasmid DNA was used to transform 40 OD$_{600}$ unit competent yeast cells by heat shock at 42 °C for 1 h. More than $10^6$ independent yeast clones were obtained and amplified on SC-U plates for 4 days before being collected for screening. The CAD-EGII strain contained an empty pRS416 plasmid was used as the control strain. The immunostaining assay and flow cytometry analysis were performed as described previously[20]. Briefly, $2.5 \times 10^6$ cells were collected at the mid-log phase and washed with PBS containing 0.5% (w/v) bovine serum albumin (BSA). For primary and secondary staining, cells were incubated statically in 20 μl of PBS with 0.5% BSA and 0.2 μl of antibodies for 1 h at 4 °C. The primary and secondary antibodies were monoclonal mouse anti-histidine tag antibody (Bio-Rad, Raleigh, NC; catalogue # MCA1396GA; clone # AD1.1.10) and goat anti-mouse IgG (H + L) secondary antibody, Biotin-XX conjugate (Thermo Fisher Scientific, Rockford, IL; catalogue # B-2763), respectively. The levels of biotin on the cell surface were quantified using Streptavidin, R-phycoerythrin (PE) conjugate (Thermo Fisher Scientific; catalogue # S866), 0.2 μl of which was added to 20 μl of cell-containing PBS with 0.5% BSA for a 30 min incubation at 4 °C in dark. Cells were washed using PBS with 0.5% BSA before switching staining reagents or flow cytometry analysis. The PE fluorescence was analysed using a LSR II Flow Cytometer (BD Biosciences, San Jose, CA). FACS experiments were performed on a BD FACS Aria III cell sorting system (BD Biosciences). In the first round of sorting, 30,000 cells representing the top 1% brightest fluorescence were collected and grown for 2 days in the SC-U medium. Then, the second round of sorting collected 93 individual yeast cells with the top 1% brightest fluorescence into a 96-well microplate. After retransformation in the CAD-EGII strain with a fresh background, the plasmids that still conferred an enhanced PE fluorescence were sent for DNA sequencing analysis. Three biological replicates of the retransformed mutants and the control strain were analysed using the immunostaining assay to estimate surface display levels by PE fluorescence.

The retransformed yeast mutants were further analysed by the CMCase assay. Briefly, 5 OD$_{600}$ unit yeast cells from overnight culture in the SC-U medium were washed twice with ddH$_2$O and resuspend in 1% (w/v) carboxymethyl cellulose (CMC) solution (0.1 M sodium acetate, pH = 5). After incubation at 30 °C for 16 h with shaking (250 r.p.m.), the supernatant was used for reducing sugar analysis by a modified DNS method[48]. The released reducing sugar amount from the CMC hydrolysis by EGII was used to quantify the enzyme activity.

**Screening of isobutanol-overproducing mutants.** Plasmid DNA of the modulation part library constructed on pRS416 was used to transform the CAD strain following the same transformation procedure as the EGII-displaying library. Collected from the SC-U plates, the strain library was used for enrichment in the synthetic SC-U medium containing increasing concentrations of norvaline (Supplementary Table 2). The CAD strain containing an empty pRS416 plasmid was used as the control strain. Serial subculturing was performed according to Supplementary Table 2, where 1% inoculum was transferred into fresh selective media when the OD$_{600}$ exceeded 2.5 in a previous round. The initial OD$_{600}$ was 0.01 for both the library and the control strain. After enrichment, single colonies were obtained by streaking the library onto a SC-U plate containing $6 \, g \, l^{-1}$ norvaline. Twenty randomly picked colonies were examined for isobutanol

production in 14 ml sealed falcon tubes with 3 ml of the SC-U medium without norvaline. After retransformation in the CAD strain with a fresh background, plasmids conferring improved isobutanol production than the control strain were sent for DNA sequencing analysis. The retransformed yeast mutants were compared with the control strain for isobutanol production. Overnight culture in the SC-U 2% glucose medium was used to inoculate 10 ml of the SC-U 2% glucose medium in 125 ml un-baffled flasks. The agitation was set to 100 r.p.m. for oxygen-limited fermentation. The initial $OD_{600}$ was adjusted to 1, and 1 ml of culture was taken at 24 h intervals to measure the cell density, ethanol titre and isobutanol titre. Metabolite quantification was performed using an Agilent 7890 gas chromatograph (GC) equipped with an Agilent 5975 mass selective detector (Agilent Inc., Palo Alto, CA) at the Roy J. Carver Metabolomics Center (University of Illinois, Urbana, IL). A previously reported protocol was used for GC program and data analysis[49].

**Screening of fast glycerol-utilizing mutants.** The same strain library used for isobutanol production screening was used for enrichment in the synthetic SC-U 3% glycerol medium. The CAD strain containing an empty pRS416 plasmid was used as the control strain. Serial subculturing was performed according to Supplementary Table 2, where 1% or 0.1% inoculum was transferred into the fresh SC-U glycerol medium when the cell density exceeded $OD_{600} = 1$ in a previous round. The initial $OD_{600}$ was adjusted to 0.01 for both the library and the control strain.

Following enrichment, 1 ml of cell suspension was used to isolate yeast plasmid DNA. Modulation parts were PCR-amplified using PRO index primer F (806rcbc756) and PRO universal primer R, as well as TER index primer F (806rcbc756) and TER universal primer R. PCR products were subjected to NGS analysis as detailed below. Reads aligned to the same modulation mode of the same gene were counted, and frequency was calculated as = (read count of individual part + 0.5)/(total read count of all parts of the same modulation mode). The same formula was also used to calculate frequencies in the plasmid library before glycerol enrichment. The modulation modes were determined based on the specific configuration of the promoter or terminator sequences, the 15 bp adaptor sequences and the directions of the cDNA inserts (Fig. 1b; Supplementary Table 5). The frequency data were calculated based on Supplementary Data 1, and results are listed in Supplementary Data 2 and plotted in Fig. 2e.

After enrichment, single colonies were obtained by streaking the library onto a SC-U glycerol plate. Twenty randomly picked colonies were examined for growth on glycerol in liquid media. After retransformation in the CAD strain with a fresh background, plasmids still conferring improved glycerol utilization relative to the control strain were sent for DNA sequencing analysis. The retransformed yeast mutants were compared with the control strain for glycerol utilization via an aerobic growth assay. Briefly, overnight culture in the SC-L 2% glucose medium was used to inoculate 3 ml of the SC-L 3% glycerol medium in 14 ml falcon tubes. The initial cell density was set to $OD_{600} = 0.1$, and cell growth was monitored at indicated time intervals.

**Estimation of gene-silencing efficiency.** To examine gene-silencing efficiency of isolated knockdown constructs, real-time qPCR was performed. The total RNAs were isolated from the mid-log phase cell cultures in the SC-U 2% glucose medium of the mutant strains and the control strain using the RNeasy mini kit (Qiagen). The cDNA synthesis was performed by the Transcriptor First Strand cDNA Synthesis kit (Roche, Indianapolis, IN). In each reaction, 5 μg of the total RNAs was used as template, and the gene-specific reverse primers for the target genes and the internal control gene *ACT1* were added in the same reaction. For control samples, only the reverse transcriptase was omitted. The primers for the cDNA synthesis were named as 'gene-name Rev' (Supplementary Table 5). qPCR reaction and data analysis were performed in a Taqman ABI 7900 real-time PCR machine using Applied Biosystems Power SYBR Green PCR Master Mix (Life Technology, Grand Island, NY) following the manufacturer's instructions. Primers for qPCR reactions were named as 'qPCR gene-name For' and 'qPCR gene-name Rev' (Supplementary Table 5).

**Manual CRISPR-assisted δ Integration.** Integration of the Cas9 gene was performed with the help of the pRS405 plasmid. Briefly, the $P_{TEF1}$-Cas9-$T_{ADH2}$ or $P_{ADH2}$-Cas9-$T_{ADH2}$ cassettes from a previous study[30] were sub-cloned into the multiple cloning site of the pRS405 plasmid. The recombinant pRS405 plasmids were then linearized in the middle of the *LEU2* marker, and transformed into the CAD strain.

The same expression cassette of modulation part libraries, $P_{TEF1}$-$T_{PGK1}$, was used in the integration donor, hence construction of the donor plasmid library followed the same procedure. Briefly, the GFP donor plasmid was converted into a backbone vector for construction of a modulation part library by replacing the *gfp* gene with the *ccdB* gene flanked by two 15 bp adaptor sequences with the primers PR608/PR611.

For manual integration, the donor plasmid library was used to transform the CAD-$P_{ADH2}$-Cas9 strains by the standard LiAc/ssDNA/PEG protocol[33] under an optimized condition, where 100 μg plasmid DNA was used to transform 40 $OD_{600}$ unit competent yeast cells by heat shock at 42 °C for 1 h. More than $10^6$

independent yeast clones were obtained in each round using the optimized protocol. The yeast transformants were amplified on twenty 150 mm diameter SC-U plates for 4 days. Cells were collected from the SC-U plates, and 20 $OD_{600}$ unit cells were spread on twenty 150 mm diameter YPAG plates containing $1 \text{ g l}^{-1}$ G418. Cells were collected after 2 days, and 20 $OD_{600}$ unit cells were spread on twenty 150 mm diameter SC-L plates containing $1 \text{ g l}^{-1}$ 5-FOA (pH = 4) and allowed for growth for 3 or 4 days. Then, 20 $OD_{600}$ unit collected cells were inoculated into 20 ml of the SC-L liquid medium containing $1 \text{ g l}^{-1}$ 5-FOA (pH = 4) in a 125-ml baffled flask. After cultivation for 1 day, plasmid curation in the SC+FOA medium was repeated once, and then 10 $OD_{600}$ unit cells were used to prepare competent cells in the SC-L liquid medium for the next round. In addition, to estimate genomic coverage of integration library, genomic DNA was isolated from 1 $OD_{600}$ unit cells collected from the SC + FOA plate. Integrated cassettes were PCR-amplified using PRO index primer F (806rcbc1631) and PRO universal primer R, as well as TER index primer F (806rcbc1631) and TER universal primer R. PCR products were subjected to NGS analysis as detailed below. Alternatively, to characterize integrated parts in low throughput, integrated cassettes were PCR-amplified separately from genomic DNA of five randomly isolated yeast mutants using PRO universal primer R and TER universal primer R. The PCR product was sub-cloned into the pRS416-TEF1p-PmeI-PGK1t plasmid and transformed into E. coli. For each yeast strain, the identities of the inserted cassettes were analysed by performing Sanger sequencing on the plasmids isolated from 20 randomly picked E. coli colonies (Supplementary Data 1d). Although the list of integrated parts was not comprehensive as only a small number of plasmids were analysed, this analysis provided a rough estimation on the numbers of integrated cassettes.

The integration procedure can also be performed with liquid media with some modifications. In particular, after transformation, all the cells were transferred into 50 ml of the SC-U medium to grow for 3 days. Then, a re-inoculation step was included to minimize the presence of untransformed cells, by inoculating 20 ml of the fresh SC-U medium with 20 $OD_{600}$ unit cells. After cultivated for 1 day, 20 $OD_{600}$ unit cells were transferred into 20 ml of the YPAG medium containing $1 \text{ g l}^{-1}$ G418, and induced for 2 days. For plasmid curation, 20 $OD_{600}$ unit induced cells were resuspended into 20 ml of the SC-L medium with $1 \text{ g l}^{-1}$ 5-FOA (pH = 4) to grow until saturation. The cultivation in the SC + 5-FOA medium was repeated once, and the cells were ready for the next round of integration.

**Integration efficiency estimated using a GFP reporter.** In earlier designs, to ensure high expression level of Cas9, a modified manual integration protocol was used (Supplementary Fig. 3a-d). Specifically, competent cells were prepared using the rich medium YPAD, and 5 μg of GFP donor plasmid was transformed into the CAD-$P_{TEF1}$-Cas9 or CAD-$P_{ADH2}$-Cas9 strains. For negative control, 5 μg of an empty donor plasmid was used to transform CAD-$P_{ADH2}$-Cas9. To estimate plasmid curation efficiency and integration efficiency without Cas9 expression, 5 μg of GFP donor plasmid was transformed into the CAD strain. Transformants were selected in YPAD containing $1 \text{ g l}^{-1}$ G418, and then automatically induced in the same medium when glucose was depleted. Cells were cultivation in YPAD for 4 days before plasmid curation. Although high integration efficiency can be achieved (Supplementary Fig. 3c,d), very few transformants can be obtained, possibly due to cellular toxicity caused by high Cas9 expression. To obtain a high transformation efficiency, the normal manual integration protocol was used, where the synthetic dropout medium was used for competent cell preparation and transformant selection, and induction was performed as a separate step (Supplementary Fig. 3e-j). Different carbon sources, including 2% ethanol, 2% glucose and 2% galactose, were used in the induction YPA medium. For the second round of integration, YPAG induction was employed. The CAD-$P_{ADH2}$-Cas9 strain treated with the GFP donor and YPAG induction in the first round was used as a new parent. The CAD strain treated with the GFP donor and YPAG induction for two rounds of integration was used as a control. After the first round of integration, 20 individual clones were randomly picked after streaking the cell population on an agar plate. To obtain mutant strains with high GFP fluorescence, FACS was performed to sort single cells with the top 0.5% fluorescence after the first round of integration into a 96-well plate using a BD FACS Aria III cell sorting system (BD Biosciences). In all cases, after plasmid curation, GFP fluorescence was analysed using a LSR II Flow Cytometer (BD, Biosciences). Gates were selected to quantify percentile of GFP-positive population, whereby cell autofluorescence was monitored using a PE channel (Supplementary Fig. 3). To analyse the genomic integration copy numbers, qPCR reactions were performed using the genomic DNA of mutant strains as template. Experiments and data analysis were performed on a Taqman ABI 7900 real-time PCR machine using the Power SYBR Green PCR Master Mix (Life Technologies) following the manufacturer's instructions. The *ALG9* gene was chosen as the reference gene (one copy on the chromosome), and the integration copy numbers of the GFP cassettes were calculated using the $2^{-\Delta\Delta Ct}$ method[50]. Primers for qPCR were named as 'qPCR gene-name For' and 'qPCR gene-name Rev' (Supplementary Table 5).

**Improving glycerol utilization using the manual protocol.** Three rounds of CRISPR-assisted δ integration with genome-wide modulation part libraries were performed on solid media with the CAD-$P_{ADH2}$-Cas9 strain. Two screening schemes were used. First, after plasmid curation in each round, 10 $OD_{600}$ unit cells

were screened on twenty 150 mm diameter SC-L plates containing 3% glycerol as the sole carbon source. Twenty colonies with remarkably bigger sizes than control (treated with an empty donor plasmid) were selected as parent for the next round of integration. The G1* mutant was identified this way. In the other scheme, three rounds of integration were performed continuously, and 10 $OD_{600}$ unit cells after plasmid curation in the third round were screened on twenty 150 mm diameter SC-L plates containing 3% glycerol as the sole carbon source. The G6 mutant was identified as one of the clones with remarkably bigger colony sizes than control. Yeast mutants were compared with the control strain (CAD-$P_{ADH2}$-Cas9) for glycerol utilization via an aerobic growth assay discussed above.

**Automated multiplex genome-scale engineering using iBioFAB.** Detailed configurations and operations of iBioFAB are described elsewhere[32]. Briefly, for hardware (Fig. 4a), iBioFAB consists of a F5 robotic arm on a 5-m track (Fanuc, Oshino-mura, Japan), an Evo200 liquid handling robot (Tecan, Männedorf, Switzerland), two shaking temperature controlled blocks (Thermo Scientific, Waltham, MA), a M1000 microplate reader (Tecan), a Cytomat 6000 incubator (Thermo Scientific), two Cytomat 2C shaking incubators (Thermo Scientific), three Multidrop Combi reagent dispensers (Thermo Scientific), four Trobot thermocyclers (Biometra, Göttingen, Germany), Vspin plate centrifuge (Agilent, Santa Clara, CA), a storage carousel (Thermo Scientific), a de-lidding station (Thermo Scientific), an Alps plate sealer (Thermo Scientific), a WASP plate sealer (Thermo Scientific), a Xpeel seal pealer (Brooks, Chelmsford, MA) and a label printer (Agilent). For software, Momentum (Thermo Scientific) was used to communicate with the peripheral devices, control the central robotic arm and program process modules. Process modules defined the unit operations and sample transportation routes between unit operations (Fig. 4b)[32]. Freedom Evoware (Tecan) was used to control the liquid handling robot and program pipetting modules. Pipetting modules specifically defined the general procedure of pipetting on the liquid handling robot, such as labware fetching from the central robotic arm, reagent dispensing and temperature control. A customized high-level modular programming environment, iScheduler[32], validated and executed process modules by sending commands in Extensible Markup Language to Momentum.

To create yeast strain libraries for identifying mutants with improved HAc tolerance, the protocol in the 'Manual CRISPR-assisted δ integration' session (Fig. 3a) was implemented on iBioFAB with some modifications (Fig. 4c). The following reagent amounts were provided for each well of a 96-well plate unless indicated otherwise. For competent cell preparation, 30 μl of overnight seed culture in SC-L was inoculated into 1 ml of SC-U on a whole 2 ml 96-well plate (Thermo Scientific), and cell growth was performed at 30 °C and 1,100 r.p.m. for 6 h. Following washing with 1 ml of ddH₂O for once and 1 ml of 0.1 M LiAc for two times, cells were transferred into a 96-well PCR plate (Thermo Scientific). One microgram of donor plasmid DNA diluted to 18.6 μl and 150.4 μl of transformation master mix[33] was added, followed by incubation at 42 °C for 1 h. After removal of transformation mix, cells were transferred into 1 ml of SC-U in a 2 ml 96-well microtiter plate. This plate normally yielded ∼10⁶ independent clones, estimated by colony-forming unit counting. For subculturing, induction and plasmid curation (Figs 3a and 4c), 100 μl of cells from a previous step was added into 900 μl of new media. Using this inoculation ratio, ∼10⁹ cells (50 $OD_{600}$ unit) were transferred between plates. Before enrichment in HAc-containing media, cells were pooled by combining 100 μl of cells from each well of a SC + FOA microtiter plate. Then, 50 μl of pooled cells was added into 1 ml of HAc-containing SC-L media in each well of a screening plate. Sixteen wells were used for each HAc concentration to screen ∼10⁷ cells (3 $OD_{600}$ unit). Cell growth was monitored by taking 20 μl of cell culture into 180 μl of ddH₂O for $OD_{600}$ measurement in each well of flat bottom 96-well assay plates (Fisher Scientific). Cells were selected with gradually increasing HAc concentrations (0.9, 1.0 and 1.0% (v/v) for the first, second and third rounds, respectively). When cell densities exceeded $OD_{600} = 0.5$ (25-fold increase with an initial $OD_{600}$ of 0.02) under a given HAc stress, cells from corresponding wells were pooled and then used to inoculate 1 ml of SC-L without HAc to prepare seed culture for the next round of genome-scale engineering. Glycerol stocks were prepared before and after screening using 500 μl of cell culture at a final glycerol concentration of 15%, and stored at − 80 °C.

**Characterization of selected HAc-resistant mutants.** After three screening cycles, selected cells from the final round were streaked onto a SC-L agar plate with 0.9% HAc (v/v; pH = 4) to obtain single colonies. Five single colonies (H1–H5) were randomly picked. For these clones, serial subculturing in non-selective SC media (pH = 4) without HAc were performed to assess genetic stability. Specifically, serial transfer was performed every 12 h at an inoculum ratio of 1% to allow five to six rounds of cell division, and >100 generations of cell division were obtained after 9 days of subculturing.

To obtain an adaptive population to HAc stress based on spontaneous mutations, the parent strain (CAD-Cas) was subjected to the same selection program used for automated strain engineering with some modifications (Supplementary Fig. 7a). No transformation and plasmid curation steps were performed between serial transfer. Due to low HAc resistance of the parent strain, an initial $OD_{600}$ value of 0.05 was used instead of 0.02 to accelerate cell growth. Screening was performed manually using 14 ml falcon tubes with 250 r.p.m. shaking, instead of automatically using 2 ml 96-well microtiter plates at 1,100 r.p.m. shaking.

Cell growth experiments in HAc media were carried out as previously described[5] at 30 °C and 250 r.p.m. Briefly, 50 μl of overnight cell cultures in SC media (pH = 4) was inoculated into 3 ml of fresh SC media to synchronize the growth phase. After 20 h, cells were transferred to 10 ml of SC media (pH = 4) with 0.9% and 1.1% (v/v) HAc in sealed 50 ml conical tubes at an initial $OD_{600}$ of 0.05. For the adaptive population, technical replicates were obtained using inoculums from the same seed culture into three separate tubes. For all other strains (the parent, mutants and mutants after subculturing), biological replicates were obtained by randomly picking three single colonies after streaking glycerol frozen stocks onto agar plates.

The fermentation performance of the H5 strain and the CAD-Cas9 strain (parent) was compared using a reported protocol[5]. In brief, three biological replicates of each strain were inoculated in 3 ml of SC-L in 14 ml round-bottom culture tubes (BD Biosciences) to grow until saturation. Then, 1 ml of saturated culture was transferred into 20 ml of SC-L in 125 ml baffled shake-flasks to grow at 30 °C and 250 r.p.m. for 20 h. The stationary-phase cells were transferred into 50 ml of SC-L with 1.1% (v/v) HAc in un-baffled 250 ml shake-flasks with an initial cell density of $OD_{600} = 1$. Fermentation was carried out under an oxygen-limited condition (30 °C and 100 r.p.m.). One millilitre of samples was taken at indicated time intervals for high-performance liquid chromatography analysis. An HPX-87H column (Bio-Rad, Hercules, CA) coupled with a refractive index detector (Shimadzu Scientific Instruments, Columbia, MD) was used to separate and analyse the concentrations of glucose and ethanol following the manufacturer's instructions.

**NGS analysis.** The shotgun genomic libraries were prepared with the Hyper Library construction kit from Kapa Biosystems (Wilmington, MA). The PCR samples were amplified with custom primers (Supplementary Fig. 9). The libraries were pooled in equimolar concentration and the pool was quantitated by qPCR and sequenced on one lane for 161 cycles on a HiSeq2500 using a TruSeq SBS sequencing kit version 4. Fastq files were generated and demultiplexed with the bcl2fastq v1.8.4 Conversion Software (Illumina, San Diego, CA).

For whole-genome shotgun data analysis, all raw reads from the four samples (WT, G1*, G6 and H5) were processed using trimmomatic (version 0.32) program[51] in single-end mode to trim adaptor sequences and low-quality bases. After the trimming of reads FastQC (version 0.11.2)[52] was run to check quality scores, and all the trimmed reads contained only bases with quality scores >30. Trimmed reads were aligned against the reference genome of CEN.PK113-7D (Saccharomyces Genome Database, http://downloads.yeastgenome.org/sequence/strains/CEN.PK/CEN.PK113-7D/), using Bwa mem (version 0.7.12)[53] with default parameters. To visualize coverage of reads aligned across the reference genomes using Integrative Genomics Viewer[54], bigwig files[55] and tile data files[54] were generated for all alignments. Read alignments in BAM format were first converted to bedGraph format using bedtools/genomeCoverageBed (version 2.21.0)[56] with parameters -bga. BedGraph files were then converted to bigWigs using bedGraphToBigWig of kent tools[55]. Tile data files were generated by running igvtools (version 2.1.24)[57] in count mode with parameters -z 7 and -w 25. From each bigwig file, a custom Perl script was used to generate average coverage information for each feature of the reference genome annotation file (in GTF format), along with s.d., coverage compared to the sequence and coverage compared to the entire genome. To visualize whether there are large rearrangements in the mutant genomes, a different S. cerevisiae genome sequence from the S288C strain was used as the reference genome (Saccharomyces Genome Database, http://downloads.yeastgenome.org/sequence/S288C_reference/genome_releases/S288C_reference_genome_R64-2-1_20150113.tgz). The S288C genome contains 16 complete chromosomal contigs, whereas CEN.PK113-7D genome contains 389 contigs. Therefore, it is more convenient to note large-scale variations of coverage using the S288C reference when visualized in Integrative Genomics Viewer. Comparing read coverage of mutant and wild-type strains, it was found that there are three large-scale genomic rearrangements in chromosome VI, VII and X of the H5 strain (Supplementary Fig. 10), while no large-scale rearrangements were observed with the G1* and G6 strains.

To examine which modulation parts were contained in a DNA sample, we PCR-amplified the modulation parts and performed NGS analysis of the resultant PCR products. Two key features are required to identify a specific modulation part: the modulation mode (overexpression or knockdown) and the target gene. The modulation mode can be determined via the presence of one of the four combinations of the promoter or terminator sequence and a 15 bp handle sequence (Supplementary Table 5). The target gene can be deduced from a cDNA fragment that is long enough for unambiguous alignment to the yeast genome. Therefore, instead of a complete modulation part, junction fragments containing promoter–15 bp handle–cDNA head or cDNA tail–15 bp handle–terminator were sequenced as they can provide enough information to identify a modulation part. To enrich those junction fragments, we used the TruSeq workflow with the following modifications: (1) during PCR amplification of modulation parts, a custom sequence ('M13F, 5′-CCAGGTTTTCCCAGTCACGAC-3′, indigo in Supplementary Fig. 9) was incorporated to one end of the PCR products, and a 12 bp barcode was included in this step to allow sample pooling in NGS (Supplementary Table 5); (2) the PCR primers annealed to immediate upstream sequences of the junctions (primer-binding site in Supplementary Table 5);

(3) after sonication and adaptor ligation using a TruSeq kit, 'PCR Primer 1.0 custom1' was used instead of the original 'PCR Primer 1.0'. Whereas the original 'PCR Primer 1.0' primer would amplify all DNA fragments by annealing to the universal adaptor sequence (brown in Supplementary Table 5), 'PCR Primer 1.0 custom1' would only amplify junction fragments by annealing to the custom 'M13F' sequence. 'PCR Primer 1.0 custom1' also contained other necessary sequences to be compatible with the TruSeq protocols (to bind flow cells, to anneal to the sequencing primer and so on).

To determine the identity of integrated modulation parts from genome sequencing results, soft-clipped reads containing one of the four combinations in Supplementary Table 5 were aligned to the reference genome to determine the target genes. Modulation modes (overexpression or knockdown) were judged by the type of combination. At least two alignments to the same gene were required for a gene to be considered a real target. A false positive list was generated using the genome sequencing results of the wild-type strain following the above analysis. Genes in the false positive list were removed from the results of the G1*, G6 and H5 strains.

Computational analysis of nearly 20 million PCR reads can be described as a de-multiplexing operation followed by read counts. Raw reads were first processed in a quality control step to remove reads with poor base quality or adaptor artifacts using Trimmomatic/0.32 (ref. 51) and Cutadapt/1.8 (ref. 58). Pooled PCR reads were first demultiplexed into different collections based on 12 bp barcodes (Supplementary Table 5) using Flexbar/2.5-beta[59], and then further demultiplexed according to four different combinations of promoter/terminator sequences and 15 bp handles (Supplementary Table 5) using Flexbar. All the sequences used in de-multiplexing were removed from reads. The remaining sequences were aligned against the reference genome of CEN.PK113-7D using Bwa mem (version 0.7.12)[53]. Aligned reads were counted depending on their location relative to gene annotations of the reference genome using the closetBed bedtools/2.24.0 (ref. 56), and divided into five different categories: downstream of a gene, upstream of a gene, start (when read and start codon intersect), middle (when read does not intersect with either codons) and end (when read and stop codon intersect). These categories are helpful to determine whether cDNA insets in the modulation parts are in full lengths. To determine coverage of the plasmid library and integrated library, a gene was considered present if at least one read was aligned. To investigate integrated modulation parts in selected cell populations from different rounds of HAc tolerance engineering, at least five reads must be found for a gene to be considered covered. In addition, a false positive list was generated via comparison between genome sequencing results and PCR library sequencing results of the H5 mutant strain. Genes in the false positive list was removed from the results from different rounds of HAc tolerance engineering. The final gene list was summarized in Supplementary Data 3.

**Gene ontology analysis.** For gene ontology enrichment analysis, genes of integrated cassettes from the first, second and third round of HAc engineering (Supplementary Data 3) were subjected to functional clustering using the MIPS functional catalogue (http://mips.helmholtz-muenchen.de/funcatDB/). Statistical tests were performed to compare frequencies of each functional class from our lists or the yeast genome. Enriched functional classes were determined as those with an associated $P$ value below 0.01 and enrichment fold above 2. Known functional classes relevant to HAc tolerance was from a previous report[60]. For glycerol utilization, genes encoded by top enriched modulation parts (Supplementary Data 2) were subjected to functional clustering following the same process as above, and functional classes with false discovery rates below 0.01 were reported (Supplementary Data 2).

**Code availability.** All computational tools used for analyses of the NGS data were available from provided references in Methods. Custom batch scripts used for execution of these computational tools can be obtained by contacting the author. All relevant data are available from the authors.

**Data availability.** The raw reads of the NGS data were deposited into the Sequence Read Archive (SRA) database (accession number: SRP072720) at the National Center for Biotechnology Information (NCBI).

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

## Acknowledgements

This work was supported by the Roy J. Carver Charitable Trust, Carl R. Woese Institute for Genomic Biology at the University of Illinois at Urbana-Champaign (UIUC), Defence Advanced Research Program Agency and National Academies Keck Futures Initiative on Synthetic Biology. T.S. acknowledges postdoctoral fellowship support from Carl R. Woese Institute for Genomic Biology (UIUC). R.C. acknowledges fellowship support from 3M Corporation. We also thank Alvaro G. Hernandez and Chris L. Wright for help with NGS, Barbara Pilas for help with FACS, and Alexander Ulanov for help with GC at Roy J. Carver Biotechnology Center (UIUC). We acknowledge Christopher J. Fields, Jenny Drnevich, Ravi Kiran Donthu and Gloria Rendon at High-Performance Biological Computing Center (UIUC) for their help with computational analysis with the NGS data.

## Author contributions

T.S. and H.Z. designed and conceived the study. T.S. devised the process for automated genome-scale engineering in yeast, and conducted strain characterization. R.C. developed the iBioFAB system, and automated the strain engineering workflow. Y.M., Y.W. and W.R. helped with DNA and genetic manipulations for creating and screening strain libraries. H.Z. supervised the research. T.S., R.C. and H.Z. wrote the manuscript.

## Additional information

**Competing interests:** The authors declare no competing financial interests.

