## [Peer Review File · Nature Communications]

Reviewer #1 (Remarks to the Author)

This manuscript showed several directed evolution methods using cDNA library for budding yeast *Saccharomyces cerevisiae*. First, the authors generated the yeast library harboring expression plasmids for sense and anti-sense cDNAs, and screened the cells that have improved cellulase display, isobutanol production and glycerol utilization. Further, they showed multiplex introduction of sense and anti-sense cDNAs into yeast by CRISPR-Cas-assisted δ -integration method. Finally, they utilized their own foundry system to demonstrate the automated multiplex genome engineering.

I think this manuscript includes several interest techniques and hot topics in the genome engineering and metabolic engineering fields. However, I have several questions for the data and conclusion (title) as follows.

1. The authors mentioned that randomly picked strains possessed 3~10 gene expression cassettes on the genome by using CRISPR-assisted δ -integration (Sup Table 1d). However, it seems that the distribution of green fluorescence was quite uniform on the dot plots of flow cytometry (Sup Fig.3). Are the gene copy-numbers multiple in the GFP integrated strains using same method?

2. The authors included the word "automated" in the title. However, I think the demonstration of "automation" is not sufficient to claim it in the current version manuscript. I think they should add the demonstration of "automated" experiments or delete the word in the title.

Reviewer #2 (Remarks to the Author)

The authors report on an approach they term multiplex genome engineering for yeast. The central premise in this work is the cloning of cDNA elements into two host vectors (equipped with delta site homology) such that overexpression and knockdown can be enabled. Certainly, "genome editing" was allowed through Cas9, but this is really a way to increase the frequency of inserting in the correct site.

The impact of the paper, methods, results, and approach are incremental at best compared to other methods like MAGE, YOGI, TRMR, etc that have been published in the literature.

It should be stated that the work only presents a nice example of automated strain manipulation, not a genome engineering approach. In fact, the title and premise is a bit of a misnomer-it is an automated random integration approach for strain engineering. While the genome is engineered per se by integrations, it is in no way a precise genome editing and is thus a confusing term.

The authors have not adequately proven their approach. There is no comparison with random drift/adaptive evolution and their approach or for random integrations etc.. This is critical for some of the phenotypes shown. What is the phenotype attainable for acetic acid tolerance without the use of this approach and simple selection in the same conditions. The isolation of only 2 strains that are marginally improved demonstrate an inherent limitation in the strain.

The resulting strains contain high amounts of homology (from multiple integrations, the use of the same promoter, the use of repeated delta sites etc). This homology will almost certainly lead to unstable strains. There is no effort made to assess the stability of the resulting strains. Thus, while this approach may identify targets, it is not a genome editing tool for producing finalized strains.

Beyond specific comments, the abstract has a number that makes no sense. Specifically, the authors tout "combinatorial diversity of $>10^{100}$ " and discussed further in the discussion on the H5 mutant is highly misleading and inaccurate. The authors come no where near sampling this diversity. The density of cells used, the total cell volumes (even with 26 copies in a cell) comes no

where near sampling this value. All references to this need to be removed as this shows a clear misunderstanding of what is actually being done here.

Reviewers' comments:

Reviewer #1 (Remarks to the Author):

*This manuscript showed several directed evolution methods using cDNA library for budding yeast *Saccharomyces cerevisiae*. First, the authors generated the yeast library harboring expression plasmids for sense and anti-sense cDNAs, and screened the cells that have improved cellulase display, isobutanol production and glycerol utilization. Further, they showed multiplex introduction of sense and anti-sense cDNAs into yeast by CRISPR-Cas-assisted δ -integration method. Finally, they utilized their own foundry system to demonstrate the automated multiplex genome engineering.*

I think this manuscript includes several interest techniques and hot topics in the genome engineering and metabolic engineering fields. However, I have several questions for the data and conclusion (title) as follows.

1. The authors mentioned that randomly picked strains possessed 3~10 gene expression cassettes on the genome by using CRISPR-assisted δ -integration (Sup Table 1d). However, it seems that the distribution of green fluorescence was quite uniform on the dot plots of flow cytometry (Sup Fig.3). Are the gene copy-numbers multiple in the GFP integrated strains using same method?

Response: There may be two reasons underlying the seemingly uniformity in **Supplementary Fig. 3**. First, it may be due to the format of data presentation. For example, the same data in the dot plots of **Supplementary Fig. 3h** and **3i** were presented as histograms in **Fig. 3c** (Round 1 and Round 2, respectively), and the histograms clearly showed a distribution of varying GFP fluorescence of the yeast populations.

Second, as the dot plots in **Supplementary Fig. 3** represent summary results on a population level, heterogeneity in GFP fluorescence of single cells may be averaged out. To examine this hypothesis, we obtained individual clones using two ways. First, the cell population after the first round of GFP integration was streaked on an agar plate. Second, single cells with very high GFP intensities were sorted into 96-well plates using FACS, and the top 3 mutants were used for further characterization. We then examined GFP signals of these strains using flow cytometry, and observed two main results.

1. Thirteen out of twenty randomly (65%) picked colonies showed GFP fluorescence, consistent with the GFP positive percentile on the population level (70%) (**Supplementary Fig. 4a**).
2. As high as **20-fold variations** in GFP intensities were observed for the 23 isolated strains (**Supplementary Fig. 4a**), and the phenotypic variations suggest differences in copy numbers and genomic loci of GFP integration. Overlays of the flow cytometry results of selected mutants were presented in histograms and dot plots to highlight phenotypic variations.

Such observations collaborated well with the results when the modulation part libraries were used as donors. Together, these results indicate multiple integration can be achieved in individual clones, and high efficiency of integration can be achieved on the population level.

The new experiments were summarized in the new **Supplementary Fig. 4** and discussed in the main text (**Line 144-150**), and the process was described in the Method section (**Line 699-703**).

2. The authors included the word “automated” in the title. However, I think the demonstration of

“automation” is not sufficient to claim it in the current version manuscript. I think they should add the demonstration of “automated” experiments or delete the word in the title.

Response: In our original submission, because the details of biofoundry configuration and automation design were included in a separated manuscript (attached in our submission, also as **Ref 29**), we did not repeat them in our manuscript. To strength the automation part, however, we moved relevant parts from the other manuscript to this revised one, including the design and configuration of the iBioFAB system (**Line 722-740** in the Method section, and the new **Fig. 4a** and **4b**), reconfiguration of iBioFAB from an established DNA assembly workflow (**Ref 29**) to the new yeast engineering workflow (**Line 171-179**), the automated yeast transformation module developed in this work (**Line 180-186**, the new **Fig. 4c**), and the process flow diagram of the automated engineering workflow (the new **Fig. 4c** and **Line 744-766** in the Method section).

Reviewer #2 (Remarks to the Author):

The authors report on an approach they term multiplex genome engineering for yeast. The central premise in this work is the cloning of cDNA elements into two host vectors (equipped with delta site homology) such that overexpression and knockdown can be enabled. Certainly, “genome editing” was allowed through Cas9, but this is really a way to increase the frequency of inserting in the correct site.

The impact of the paper, methods, results, and approach are incremental at best compared to other methods like MAGE, YOGÉ. TRMR, etc that have been published in the literature.

Response: In our original submission, we have compared our methods and results with recombineering-based technologies (MAGE, YOGÉ and TRMR), and discussed why our work represented a significant advance in the field (**Line 211-221** and **229-238** in the original manuscript). However, our discussion was focused on different aspects of our method and specific considerations in yeast engineering. We did not center on criticism of existing technologies, as it may be unnecessary as they were not developed for yeast. We apologize if such organization in writing may create confusion in understanding the substantial advances of our methods. Here, we provided an integrated and organized criticism of existing methods for applications in yeast engineering. Most points were already covered in different paragraphs in the main text.

Technically, there are two main reasons why recombineering-based methods (MAGE, TRMR and YOGÉ) are not sufficient for automated yeast engineering. Although recombineering is efficient in bacteria, it is not in yeast as indicated by the YOGÉ method (YOGÉ showed 1% recombineering efficiency in yeast, compared with 30% using MAGE in *E. coli*, and 70% using our system in yeast). Hence, it is simply impossible to apply MAGE or TRMR in an automated workflow for *S. cerevisiae* to allow genome-wide coverage or multiplex modifications. Second, only short (90 nt) oligonucleotides can be used to introduce mutations efficiently without antibiotic selection (as in MAGE). When longer DNA cassettes were used in TRMR, antibiotic selection became necessary. This limitation on the editing scale is problematic in yeast in two ways: 1) genetic regulation is more complicated in eukaryotes, and small nucleotide changes are inefficient to modulate gene expression in yeast; 2) the use of antibiotic selection renders multiplex modifications impossible. Therefore, new methods other than recombineering must be developed for yeast.

Also, our achievement outperforms MAGE and TRMR in the general field of microbial genome-scale engineering. First, MAGE can only explore combinatorial diversity among predefined targets, while our methods can expand the scope to a genome-wide scale. Second, like TRMR in *E. coli* (95%), we achieved a genome coverage of 92% for both genetic overexpression and knockdown. But we devised a much simpler and more generic method relative to TRMR, without the requirement for microarray DNA synthesis or complicated barcoding schemes. Notably, it is the first time that both overexpression and knockdown modifications can be introduced simultaneously on a genome scale in yeast. Moreover, TRMR cannot be used to create multiplex modifications due to the use of antibiotic selection as discussed above. Third, although TRMR and MAGE were performed sequentially for multiplex mutations, combinatorial optimization was again limited in predefined pools of targets, which may miss synergistic beneficial mutations with weak phenotypes when created in isolation (**Ref 5** and **38**). Furthermore, TRMR and MAGE cannot be seamlessly combined as they use different methods of mutagenesis: TRMR inserts synthetic modulation cassettes in promoters and MAGE introduces small nucleotide changes in RBSs. On the contrary, we devised a standard and scalable way for introduction of different genetic mutations using the same integration method, which is critical to achieve automated engineering.

Still, we think it is more logic and natural to focus on our own design, instead of why existing methods are inadequate in yeast engineering, an application which they were not designed for. But we understand this may lead to confusion in why our method and results present major advances in the field. We propose the following changes to clarify potential confusion.

1. We included the values of recombination efficiencies of YOGEE, MAGE and our method for direct comparison (**Line 254-256**)
2. We included a new reference (the new **Ref 36**) to explain in details why small scale changes used in recombineering may not be sufficient for genetic modulation in yeast (**Line 256-261**). In this reference, 4 to 71 mutations randomly distributed in a range of 400 bp sequence is necessary to modify the strength of a yeast promoter efficiently.
3. We included the above integrated criticism of existing methods in yeast engineering as **Supplementary Text**, and mentioned it in the revised manuscript (**Line 256**).

It should be stated that the work only presents a nice example of automated strain manipulation, not a genome engineering approach. In fact, the title and premise is a bit of a misnomer-it is an automated random integration approach for strain engineering. While the genome is engineered per se by integrations, it is in no way a precise genome editing and is thus a confusing term.

Response: we do understand the possible confusion raised by the reviewer, as currently the popular understanding of genome engineering has been limited to genome editing. In fact, due to the exact same concern as the reviewer, in the original submission, we carefully used “genome-scale engineering” instead of “genome engineering” in the title and abstract, and across the manuscript in most cases.

However, “genome engineering” is an evolving concept. Before the recent emergence of programmable nucleases such as TALEN and CRISPR-Cas, genome engineering is not restricted to precise genome editing. In fact, genome engineering is proposed as a distinct research area from traditional genetic engineering, based on its intensive nature (at least two distinct regions of a genome) and the genome-wide scale (*Nat Biotechnol*, **27**, 1551-62 (2009); *Mol Sys Biol*, **9**, 641 (2012)), not on the nature of modifications (genome editing, rewriting of regulatory networks, or *de novo* synthesis). In fact, the use of regulatory RNAs in microbial genome engineering has gathered intensive attention recently (*Nat Rev*

Microbiol, **12**, 341–354 (2014); *Nat Biotechnol* **31**, 170-174 (2013)), and one main advantage is to modulate gene expression without modifying targeted genome loci. As one of the first groups applying both genome editing and regulatory RNAs for yeast engineering, we believe they are equally important and complementary to each other given their unique pros and cons. In the original submission, we explained the reason why we chose *trans-acting* modulation over CRISPR-based genome editing, and pointed out the need for further development of CRISPR for multiplex genome-scale engineering in yeast (**Line 238-247** in the original submission), which is currently one of the research focuses in our group.

Having said that, we agree with the reviewer' comment. In the revised manuscript, in addition to the title, abstract, and most cases in the main text, where we did use “genome-scale engineering” in the original submission, we replaced all the remaining “genome engineering” with “genome-scale engineering” when this term was used to refer to our work.

The authors have not adequately proven their approach. There is no comparison with random drift/adaptive evolution and their approach or for random integrations etc.. This is critical for some of the phenotypes shown. What is the phenotype attainable for acetic acid tolerance without the use of this approach and simple selection in the same conditions. The isolation of only 2 strains that are marginally improved demonstrate an inherent limitation in the strain.

Response: in our original submission, although there was no direct comparison between adaptive evolution and our approach, we did include results and a literature summary suggesting our method outperform traditional evolutionary engineering.

1. In the first round of selection, a control strain transformed with an empty donor plasmid were processed in parallel with the library to account for any spontaneous mutations. Our results clearly indicated that the library outperformed the control strain within the same time frame under all acetic acid concentrations (the old **Fig. 3d** and now **Fig. 5a**).
2. As the integration efficiency (70%) is less than 100%, there are always cells in the evolving population that are not modified. These cells can serve as built-in controls for random drift/adaptive evolution (the old **Line 143-145**).
3. We compiled the published literature on HAc resistance engineering in yeast using traditional methods (including adaptive evolution and rational approaches) for attainable acetic acid tolerance (**Supplementary Table S7**), and showed our isolated mutants exhibited the highest HAc tolerance levels not achieved using traditional techniques, including long-term adaptive evolution experiments. This comparison also proves that the improvement obtained using our method is not “marginal”.

Despite these existing evidences, we performed adaptive evolution using the same selection conditions (three rounds of selection in 0.8-1.1% HAc media) for more direct comparison as suggested by the reviewer. Two observations confirmed that our method greatly accelerated the occurrence of HAc resistance relative to adaptive evolution.

1. During three rounds of simple serial transfers in HAc media, although growth rates were increased with 0.8 and 0.9% HAc, no substantial growth was observed in 1.0% and 1.1% HAc media (the new **Supplementary Fig. 7a**). This was opposite to our libraries, which could grow in 1.0% and 1.1% HAc media during the second and third round of screening, respectively (now the **Supplementary Fig. 6a-d**).

2. After three rounds of adaptive evolution, the evolved population was examined for biomass accumulation in HAc media together with the parent strain and individual engineered mutants isolated after the third round of automated engineering (the new **Supplementary Fig. 7b** and **7c**). In the presence of 0.9% HAc, HAc resistance was observed in the order of mutants>adaptive population>parent. In the presence of 1.1% HAc, the growth of the parent strain and adaptive population was completely inhibited, whereas four isolated mutants showed substantial growth. Notably, no reported *S. cerevisiae* strains can ever grow in 1.1% HAc media (**Supplementary Table S7**),

These new experiments were summarized in the new **Supplementary Fig. 7a-c**, **Line 197-202** and **Line 211-217** in the main text, and **Line 776-791** in the Method section.

The resulting strains contain high amounts of homology (from multiple integrations, the use of the same promoter, the use of repeated delta sites etc). This homology will almost certainly lead to unstable strains. There is no effort made to assess the stability of the resulting strains. Thus, while this approach may identify targets, it is not a genome editing tool for producing finalized strains.

Response: Although substantially improved traits can be obtained using our method, we agree with the reviewer that our method is a discovery tool rather than an engineering tool for finalized strains. In fact, we have clearly stated in the first sentence that the main purpose of genome-scale engineering is “for large-scale genotype-phenotype mapping”, and hence not to produce finalized strains. In our opinion, given the ever-improving technologies of genome editing and synthesis, creating a microbial genome with large-scale defined modifications will no longer be a bottleneck soon. However, the major challenge is the lack of reliable design rules to achieve desirable phenotypes. To identify engineering targets and to understand underlying mechanisms for a given trait, therefore, systematic evaluation of genome-wide perturbations remains essential but challenging. In this context, we tried to devise an efficient method to create and screen multiplex genome-scale diversities in yeast. Moreover, although not stated explicitly, for future work we mainly proposed on how to understand the mechanisms conferring improved phenotypes (**Line 262-277** in the original submission), with the mindset that our method is a discovery tool.

Having said that, we still performed the stability test of isolated mutants as suggested by the reviewer. We expected that there should be no dramatic differences between our mutants and numerous yeast strains previously engineered using traditional δ -integration, as they all contain “high amounts of homology (from multiple integrations, the use of the same promoter, the use of repeated delta sites etc)”. Consistent with previous findings that δ -integration strains are generally stable with the possibility of genetic instability, 4 out of 5 isolated mutants showed no differences in HAc resistance before and after 100 generations of cell division in non-selective media, but one mutant (H5) lost the ability to grow in 1.1% HAc media (the new **Supplementary Fig. 7d**).

To avoid possible confusion raised by the reviewer, in the revised manuscript we explicitly stated that the main purpose of our method is to search and understand multiplex mutations on a genome scale, not to produce “finalized strains” (**Line 322-327**). The stability experiments were presented in the new **Supplementary Fig. 7d**, **Line 217-225** in the main text, and **Line 770-773** in the Method section.

Beyond specific comments, the abstract has a number that makes no sense. Specifically, the authors tout “combinatorial diversity of $>10^{100}$ ” and discussed further in the discussion on the H5 mutant is highly

misleading and inaccurate. The authors come no where near sampling this diversity. The density of cells used, the total cell volumes (even with 26 copies in a cell) comes no where near sampling this value. All references to this need to be removed as this shows a clear misunderstanding of what is actually being done here.

Response: we agree with the reviewer that we did not sample this diversity, as we clearly stated in **Line 254-255** of the original submission (“this diversity far exceeds the actual size of cell populations”). However, **creation** of combinatorial diversity that exceeds the actual cell numbers through successive cycles of mutagenesis in an evolving population is possible (as demonstrated in MAGE, *Nature* **460**, 894-898 (2009)), although only a subset of the theoretical diversity can be **screened** or **sampled** at any given time. Here a similar approach of iterative rounds of mutagenesis was employed, and we showed how 10^{100} was calculated. Such feature should be common to all methods using successive mutagenesis cycles to create libraries, and there should be nothing misleading.

However, we understand that the number on itself outside of proper context may be confusing, so that we deleted the specific numbers from the abstract and the discussion in the revised manuscript. But we think it is legitimate to keep the calculations to describe the gigantic diversity space for multiplex genome-scale libraries, which is one of the reasons for automation (the same logic was also used in MAGE). Moreover, we stated that such diversity was not comprehensively sampled (**Line 299-300**), in the specific way the reviewer phrased, after our original statement with essentially the same implication (“this diversity far exceeds the actual size of cell populations”).

Reviewer #1 (Remarks to the Author)

The authors has addressed my concerns properly and I feel the body of the current version manuscript is acceptable.

Reviewer #2 (Remarks to the Author)

The authors have addressed all of the points raised in the prior round of review and now present a much more complete story. The inclusion of the comparison data with adaptive evolution and further discussion of alternative genome editing tools was essential. Likewise, the inclusion of more details about the iBioFab now provide a further justification of the automation that was being discussed.

One remaining comment associated with new text that was described. The authors discuss that differences in the GFP level of selected clones after the delta integration may be due to copy number. The authors should consider explicitly measuring copy number of these genes to substantiate this claim (this is a rather straightforward and simple to run experiment).

Response to reviewer's comments

Reviewer #1 (Remarks to the Author)

The authors has addressed my concerns properly and I feel the body of the current version manuscript is acceptable.

We appreciate this reviewer's help to improve the quality of our manuscript.

Reviewer #2 (Remarks to the Author):

The authors have addressed all of the points raised in the prior round of review and now present a much more complete story. The inclusion of the comparison data with adaptive evolution and further discussion of alternative genome editing tools was essential. Likewise, the inclusion of more details about the ibioFab now provide a further justification of the automation that was being discussed.

We appreciate this reviewer's help to improve the quality of our manuscript.

One remaining comment associated with new text that was described. The authors discuss that differences in the GFP level of selected clones after the delta integration may be due to copy number. The authors should consider explicitly measuring copy number of these genes to substantiate this claim (this is a rather straightforward and simple to run experiment).

Response: As suggested by this reviewer, we performed quantitative PCR on genomic DNA samples of isolated GFP strains to measure the integration copy numbers of the GFP gene. We employed the widely used $2^{-\Delta\Delta C_t}$ method (Methods 25, 402-408 (2001)) using the *ALG1* gene as a reference. We did observe 2-9 integration copies of GFP among the isolated strains (Supplementary Fig. 4a), which is consistent with our claim in the last submission. The new data was included in Supplementary Figure 4a and discussed in the main text. Experimental procedures were described in the Methods section.